

# Bayesian estimation of the measurement of interactions in epidemiological studies

Shaowei Lin[1,*], Chanchan Hu[1,*], Zhifeng Lin[1] and Zhijian Hu[1,2]

[1] Department of Epidemiology and Health Statistics, School of Public Health, Fujian Medical University, FuZhou, Fujian, China
[2] Key Laboratory of Ministry of Education for Gastrointestinal Cancer, Fujian Medical University, FuZhou, Fujian, China
* These authors contributed equally to this work.

## ABSTRACT

**Background**. Interaction identification is important in epidemiological studies and can be detected by including a product term in the model. However, as Rothman noted, a product term in exponential models may be regarded as multiplicative rather than additive to better reflect biological interactions. Currently, the additive interaction is largely measured by the relative excess risk due to interaction (RERI), the attributable proportion due to interaction (AP), and the synergy index (S), and confidence intervals are developed via frequentist approaches. However, few studies have focused on the same issue from a Bayesian perspective. The present study aims to provide a Bayesian view of the estimation and credible intervals of the additive interaction measures.

**Methods**. Bayesian logistic regression was employed, and estimates and credible intervals were calculated from posterior samples of the RERI, AP and S. Since Bayesian inference depends only on posterior samples, it is very easy to apply this method to preventive factors. The validity of the proposed method was verified by comparing the Bayesian method with the delta and bootstrap approaches in simulation studies with example data.

**Results**. In all the simulation studies, the Bayesian estimates were very close to the corresponding true values. Due to the skewness of the interaction measures, compared with the confidence intervals of the delta method, the credible intervals of the Bayesian approach were more balanced and matched the nominal 95% level. Compared with the bootstrap method, the Bayesian method appeared to be a competitive alternative and fared better when small sample sizes were used.

**Conclusions**. The proposed Bayesian method is a competitive alternative to other methods. This approach can assist epidemiologists in detecting additive-scale interactions.

# INTRODUCTION

Interaction occurs when the impact of an independent variable (X) on a dependent variable (Y) varies at different levels of a moderating variable (Z) (*Andersson, Cuervo-Cazurra & Nielsen, 2014*). In epidemiology, identifying the interaction between two factors for disease risk is important (*Szklo , 2004*) because this interaction is significantly associated with disease prevention and intervention. For example, from the perspective of public health,

Corresponding author
Zhijian Hu, huzhijian@fjmu.edu.cn

the combined effect of smoking and asbestos on lung cancer surpasses the combination of the individual effects, and a reduction in either factor would also reduce the risk attributable to the other factor in terms of developing lung cancer (*Ngamwong et al., 2015*).

Epidemiologists frequently employ exponential models such as logistic regression and Cox regression to analyze the disease rates and risks (*Rothman, Greenland & Lash, 2008*). Under an exponential model, there are two scales of interaction: additive and multiplicative. The latter is often employed to assess interactions by including a product interaction term in an exponential model, which implies that the combined effect is larger (or smaller) than the product of the individual effects. However, Rothman and other authors (*Rothman, 1976*; *Andersson et al., 2005*; *VanderWeele & Robins, 2007*) have argued that the examination of interactions on an additive scale makes more sense than that on a multiplicative scale, in which a positive or negative interaction on an additive scale means that the effect in combination is larger or smaller than the sum of the individual effects, respectively.

*Rothman (1986)* proposed different measures to estimate interactions on an additive scale by means of relative risk, such as the relative excess risk due to interaction (RERI), the attributable proportion due to interaction (AP) and the synergy index (S). *Hosmer & Lemeshow (1992)* introduced how to use the delta method to calculate a symmetric confidence interval in a logistic regression model. Furthermore, *Knol et al. (2007)* generalized those measures as a departure from additivity in the case of continuous determinant factors. However, due to the skewness of these measures, methods of asymmetric confidence intervals, such as the bootstrap percentile confidence interval (*Assmann et al., 1996*), variance estimate recovery method (*Zou, 2008*), and profile likelihood (*Richardson & Kaufman, 2009*), have been developed and have produced competitive results (*Andersson et al., 2005*; *Kuss, Schmidt-Pokrzywniak & Stang, 2010*; *Nie et al., 2010*). In addition to frequentist methods, *Chu, Nie & Cole (2011)* described the use of the Bayesian method to estimate the RERI in a linear additive odds ratio model, which hardly worked with the AP and S.

However, for all the methods mentioned above, the estimation of interactions on an additive scale can be applied only to risk factors rather than to preventive factors (*Yang et al., 2010*; *Chatterjee, Shi & García-Closas, 2016*; *Olsson, Barcellos & Alfredsson, 2017*). When either of the two factors at discussion is not a risk factor, both factors should be considered separately or jointly by choosing the low-risk category as the reference level (*Rothman, Greenland & Lash, 2008*; *de Mutsert et al., 2009*; *Knol et al., 2011*). However, this strategy involves applying a logistic model twice. The risk level or combination of factors are first determined, after which the coefficients of the factors are estimated, with the low risk level serving as a reference. Notably, the relation of the estimated coefficients between the two logistic models can be revealed by resetting only the reference level; however, the calculation of confidence intervals becomes complicated with the troublesome involvement of covariance matrices.

This article describes how credible intervals of the three measures of interaction on an additive scale can be evaluated using the Bayesian method. Credible intervals are analogous to the in frequentist statistics (*Bolstad & Curran, 2016*), although they differ on a philosophical basis (*VanderPlas, 2014*). The Bayesian method is commonly employed

in epidemiology and other applications (*Ashby, 2006*; *MacLehose et al., 2009*; *Hamra et al., 2013*; *Ahrens & Pigeot, 2014*) due to the simplicity of Bayesian inference. Bayesian inference depends only on posterior samples of the parameters. Once a posterior sample is obtained, a posterior sample of any function of the parameters can be obtained by applying the corresponding function to the posterior sample of those parameters. A sample from the posterior distribution of coefficients of the logistic model can be generated using Markov chain Monte Carlo (MCMC) methods with a random walk Metropolis algorithm (*Bolstad & Curran, 2016*). Once a sample is obtained, regardless of whether the logistic model includes preventive factors or risk factors, the posterior samples of the three measures of interaction on an additive scale can be calculated directly by applying the corresponding functions. If there are preventive factors, the functions are composed of two parts: the coefficients corresponding to the new reference (Appendix S1A) and the measure's functions. Thus, the logistic model requires only one execution. Based on the posterior samples, credible intervals, such as the highest posterior density interval or equal-tailed interval, can easily be assessed.

## Measures of additive interaction

Consider two binary factors, A and B, with values of 0 and 1, where 0 denotes the absence of a factor and 1 denotes its presence. Four possible combinations exist: $A_0B_0$, $A_1B_0$, $A_0B_1$ and $A_1B_1$. Suppose the reference group is $A_0B_0$, and let $RR_{10}$, $RR_{01}$, and $RR_{11}$ denote the relative risks for groups $A_1B_0$, $A_0B_1$ and $A_1B_1$, respectively. *Rothman, Greenland & Lash (2008)* developed three measures of interaction on an additive scale: the relative excess risk due to interaction (RERI),

$$RERI = RR_{11} - RR_{10} - RR_{01} + 1,$$

the attributable proportion due to interaction (AP),

$$AP = \frac{RERI}{RR_{11}} = \frac{RR_{11} - RR_{10} - RR_{01} + 1}{RR_{11}},$$

and the synergy index (S)

$$S = \frac{RR_{11} - 1}{(RR_{10} - 1) + (RR_{01} - 1)} = \frac{RR_{11} - 1}{RR_{10} + RR_{01} - 2}.$$

Note that both the RERI and AP can range from $-\infty$ to $+\infty$, while S can range from 0 to $+\infty$. If no interaction is present on the additive scale, both the RERI and AP will be equal to 0, and S will be equal to 1; if there is more than additivity, both the RERI and AP will be greater than 0, and S will be greater than 1; if there is less than additivity, both the RERI and AP will be less than 0, and S will be less than 1.

The RR is readily available in cohort studies. However, in case-control studies, the RR can be approximated by the odds ratio (OR) if the prevalence is sufficiently rare (Appendix S1B). As suggested by *Rothman, Greenland & Lash (2008)*, the RERI, AP and S can be estimated *via* multiple logistic regression with indicator variables created for categories $A_1B_0$, $A_0B_1$ and $A_1B_1$. Given that

$$logit(p) = \log(\frac{p}{1-p}) = \beta_0 + \beta_1 I(A_1B_0) + \beta_2 I(A_0B_1) + \beta_3 I(A_1B_1) + X\gamma \qquad (1)$$

where $p = P(case|A, B, X)$ is the corresponding probability of the case given factors A, B and X (X is a vector of the potential confounders) and $I(\cdot)$ is the indicator function; then, $OR_{10} = e^{\beta_1}$, $OR_{01} = e^{\beta_2}$ and $OR_{11} = e^{\beta_3}$ can be substituted for the RRs in the three measures of interaction. As noted by *Hosmer Jr, Lemeshow & Sturdivant (2013)*, the ORs can be represented as $OR_{10} = e^{\eta_1}$, $OR_{01} = e^{\eta_2}$ and $OR_{11} = e^{\eta_1 + \eta_2 + \eta_3}$ based on logistic regression with factors A and B and the product of A and B

$$logit(p) = \eta_0 + \eta_1 A + \eta_2 B + \eta_3 AB + X\gamma. \tag{2}$$

Because both models are saturated, the two estimations are equivalent.

## Methods for calculating confidence intervals

The methods used to calculate confidence intervals fall into two categories: symmetric intervals based on a normal distribution and asymmetric intervals based on quantile estimation. Let lnS denote S on the log scale and Z denote any one of the three measures RERI, AP or lnS. As proposed by Hosmer and Lemeshow, the confidence interval of Z is obtained by assuming a normal distribution for Z. The variance in Z can be estimated using the delta method, which employs the first-order approximation of a Taylor series (*Hosmer & Lemeshow, 1992*). Therefore, the 95% confidence interval of the delta method for Z, which is symmetric about the point estimate, is given by $\hat{Z} \pm 1.96 \times \sqrt{\hat{D}(Z)}$, where $\hat{Z}$ and $\hat{D}(Z)$ are the point estimate and the variance estimate for Z, respectively. The confidence intervals for the RERI and AP can be obtained directly, while the confidence interval for S can be obtained using an exponential function. *Zou (2008)* suggested a recover variance estimate of the variance for measures that considers Taylor series expansion using the multivariate delta method.

Another way to estimate confidence intervals is through bootstrapping (*Efron & Tibshirani, 1994*). This technique allows estimation of the sampling distribution of almost any statistic using random sampling, as do the measures of additive interaction. Let Z denote any one of the three measures RERI, AP or S. The bootstrap samples are resampled from the original sample, and then Z can be estimated in each of the bootstrap samples. When obtaining the distribution of Z, deriving estimates of standard errors and confidence intervals is straightforward. There are several methods for constructing confidence intervals from the bootstrap distribution.

The first of these methods is the normal bootstrap method. The method assumes that Z follows a normal distribution. The variance $\hat{D}(Z)$ is estimated from the bootstrap samples, and the confidence interval is calculated as $\hat{Z} \pm 1.96 \times \sqrt{\hat{D}(Z)}$ Alternatively, the quantile method is the most popular method for constructing confidence intervals. Let $Z_\alpha^*$ denote the $\alpha$ percentile of the bootstrap-estimated distribution for Z. *Efron & Tibshirani (1994)* used percentiles of the bootstrap distribution to estimate the $1 - \alpha$ confidence interval for Z as (percentile bootstrap, PB)

$$(Z_{\alpha/2}^*, Z_{1-\alpha/2}^*).$$

*Davison & Hinkley (1997)* proceeded in a similar way, but a different formula was used to construct the confidence interval for Z, as (basic bootstrap, BB)

$$(2\hat{Z} - Z_{1-\alpha/2}^*, 2\hat{Z} - Z_{\alpha/2}^*).$$

## Bayesian estimation of the additive measure

Given a model, there are three steps involved in Bayesian analysis (*Bolstad & Curran, 2016*): (1) determining the marginal likelihood of the data, (2) specifying prior probabilities for the parameters, and (3) applying Bayes' theorem to predictor variables and the outcome variable *via* Bayesian modeling.

Let Y denote the binary outcome variable. Consider the logistic regression of the form

$$Y : Bernoulli(p)$$

where *Bernoulli*(·) is a Bernoulli distribution and $p$ satisfies Eq. (1) or Eq. (2). The likelihood for the logistic regression is

$$p(x|\theta) = \prod_i h(x;\theta)^Y (1 - h(x;\theta))^{1-Y}$$

where $\theta$ is a parameter (such as $\theta = (\beta_0, \beta_1, \beta_2, \beta_3, \gamma)$ in Eq. (1)) and $h(x;\theta) = \frac{1}{1+e^{-logit(p)}}$ is the predicted probability that Y is 1.

The prior probability is an unconditional probability that is assigned before any data are accounted for. There are three main categories of prior distributions: informative priors, weakly informative priors and noninformative priors. When a family of conjugate priors exists, the conjugate priors are chosen for computational efficiency. However, there is no conjugate prior for the likelihood function in logistic regression. When Bayesian inference is performed analytically, this makes the posterior distribution difficult to calculate except in very low dimensions. However, software such as OpenBUGS (*Lunn et al., 2009*), JAGS (*Plummer, 2003*) and Stan (*Carpenter et al., 2017*) allows these posteriors to be computed *via* simulation; hence, a lack of conjugacy is not a concern. This article employs noninformative or weakly informative priors $p(\theta)$ for all model parameters.

According to Bayes' theorem, the joint posterior distribution of the model parameters is proportional to the product of the likelihood and priors,

$$p(\theta|x) \propto p(\theta) \cdot p(x|\theta) = p(\theta) \prod_i h(x;\theta)^Y (1 - h(x;\theta))^{1-Y}.$$

This equation is too complex to solve analytically; therefore, Monte Carlo methods are often used to summarize the posterior distribution. A combination of Monte Carlo and Markov chain can result in effective sampling and evaluation. Thus, this posterior computation can be accomplished easily using Markov chain Monte Carlo (MCMC) methods with a random walk Metropolis algorithm. Since RERI, AP and S are functions of $\theta$, the values of the three measures of interaction can be calculated directly when a sample is generated from the posterior distribution $p(\theta|x)$.

The uncertainties of the RERI, AP and S can be summarized by giving a range of values on the posterior probability distribution that includes 95% of the probability, $(P_{2.5}, P_{97.5})$, which is called a 95% credibility interval (calculated based on an equal-tailed interval). Similarly, in a Bayesian credibility interval, the null hypothesis is rejected if the credibility interval does not encompass the parameters of the null hypothesis. For example, there is an

interaction if the credibility interval of an RERI does not contain 0. Notably, the confidence interval captures the uncertainty about the interval obtained (whether it contains the true value or not), and it cannot be interpreted as a probabilistic statement about the true parameter value. However, the credible interval captures the current uncertainty in the location of the parameter value and thus can be interpreted as a probabilistic statement about the parameter.

## Preventive factors

Measures of interaction were developed for risk factors, so it is not appropriate to use them directly when a preventive factor exists. The recoded method, which can be seen as a choice of reference category, is used to address this issue (*Knol et al., 2011*). The use of the recoded method and the use of an updated model is a cumbersome and tedious process. In fact, when the reference category changes, the new logistic regression coefficients $\theta' = (\beta_0', \beta_1', \beta_2', \beta_3')$ are completely dependent on the original logistic coefficients $\theta = (\beta_0, \beta_1, \beta_2, \beta_3)$, as resetting the reference category is equivalent to subtracting the new reference category coefficient from $\theta$ (*Hosmer Jr, Lemeshow & Sturdivant, 2013*). The relationships between $\theta'$ and $\theta$ for the three interaction measures are shown in Appendix S1A. In the Bayesian approach, the three measures of interaction are calculated directly by setting the reference category as that whose coefficient is the smallest (set $\beta_0 = 0$ in the dummy coding scheme). Thus, it is not necessary to update the model, and the uncertainty of the parameters will be obtained from the posterior sample. The R (*R Core Team, 2023*) function for computing new coefficients based on the lowest risk category as a reference category is provided in Appendix S1C.

## SIMULATION STUDY SETTING

To assess the performance of Bayesian estimation of the additive measure, simulation studies were conducted to compare the results of confidence interval estimation by these methods. Regardless of the effect measure used, OR, RR, or hazard ratio, the evaluation process was similar, so a case-control design was employed. The values of $OR_{01}$, $OR_{10}$ and $OR_{11}$ were set as in the study by *Assmann et al. (1996)*, where 20 scenarios (2 $OR_{01} \times$ 2 $OR_{10} \times$ 5 $OR_{11}$) are described in Table 1. These scenarios include situations with strong synergy, weak synergy, no interaction, weak antagonism or strong antagonism.

In each scenario, the sample size was set to $n = 600$, and the factor exposure rate was fixed, in which the proportions of subjects exposed to factor A alone, factor B alone, and both factors A and B were 0.1, 0.2 and 0.1, respectively. In addition to the parameters mentioned above, two additional parameters were considered: the proportion of the case number relative to the sample size, which changes from $p_1 = 1/2$ to $p_2 = 1/3$, *i.e.*, the number of cases is reduced from 300 to 200; and the other is where there are potential confounders X and the coefficient of the variable is set to $\gamma = 0.5$ when containing a confounding variable.

Three simulation studies were designed to evaluate the effects of the methods: balanced design ($p_1 = 1/2$), unbalanced design ($p_2 = 1/3$) and balanced design with a confounding variable. For each combination of parameters in each study, 1,000 replicates were performed. Bayesian estimation was compared with the delta method and bootstrap

**Table 1  The ORs setting for simulation.**

| Scenario | $OR_{01}$ | $OR_{10}$ | $OR_{11}$ | RERI | AP | S |
|---|---|---|---|---|---|---|
| A1 | 4.0 | 5.0 | 20.000 | 12.00 | 0.60 | 2.71 |
| A2 | 4.0 | 5.0 | 12.000 | 4.00 | 0.33 | 1.57 |
| A3 | 4.0 | 5.0 | 8.000 | 0.00 | 0.00 | 1.00 |
| A4 | 4.0 | 5.0 | 6.000 | −2.00 | −0.33 | 0.71 |
| A5 | 4.0 | 5.0 | 4.000 | −4.00 | −1.00 | 0.43 |
| B1 | 2.0 | 5.0 | 15.000 | 9.00 | 0.60 | 2.80 |
| B2 | 2.0 | 5.0 | 9.000 | 3.00 | 0.33 | 1.60 |
| B3 | 2.0 | 5.0 | 6.000 | 0.00 | 0.00 | 1.00 |
| B4 | 2.0 | 5.0 | 4.500 | −1.50 | −0.33 | 0.70 |
| B5 | 2.0 | 5.0 | 3.000 | −3.00 | −1.00 | 0.40 |
| C1 | 4.0 | 2.5 | 13.750 | 8.25 | 0.60 | 2.83 |
| C2 | 4.0 | 2.5 | 8.250 | 2.75 | 0.33 | 1.61 |
| C3 | 4.0 | 2.5 | 5.500 | 0.00 | 0.00 | 1.00 |
| C4 | 4.0 | 2.5 | 4.125 | −1.38 | −0.33 | 0.69 |
| C5 | 4.0 | 2.5 | 2.750 | −2.75 | −1.00 | 0.39 |
| D1 | 2.0 | 2.5 | 8.750 | 5.25 | 0.60 | 3.10 |
| D2 | 2.0 | 2.5 | 5.250 | 1.75 | 0.33 | 1.70 |
| D3 | 2.0 | 2.5 | 3.500 | 0.00 | 0.00 | 1.00 |
| D4 | 2.0 | 2.5 | 2.625 | −0.88 | −0.33 | 0.65 |
| D5 | 2.0 | 2.5 | 1.750 | −1.75 | −1.00 | 0.30 |

method. As suggested by *Assmann et al. (1996)* due to the skewed distribution of the three measures of interaction, the percentile bootstrap was suitable. Thus, only the percentile bootstrap was performed in the simulation studies. The bootstrap sampling, with 1,000 resamples, was performed separately within cases and controls. Therefore, the number of cases and controls in the bootstrap samples were identical to those in the original sample. Therefore, for each sample, point estimates of the interaction measures were calculated, as were 95% confidence intervals for the delta method and percentile bootstrap method and 95% credible intervals for the Bayesian method.

The posterior distributions of the parameters were estimated *via* the MCMC method with a random walk Metropolis algorithm, which can be performed easily *via* the R package MCMCpack (*Martin, Quinn & Park, 2011*). Specifically, the Markov chain was run for a total of 20,000 iterations with 10,000 burn-in iterations. To monitor convergence, multiple chains with different initial values were constructed, and trace plots were examined visually. The convergence of Markov chains was further assessed using the standard convergence statistic. Because an improper prior distribution may lead to inaccurate posterior estimates, proper but diffuse prior distributions with weak information for the parameters were employed (*Spiegelhalter & Richardson, 1995*). Alternatively, noninformative priors, such as the Jeffreys prior or uniform prior with proper bounds, may also be used (*Bolstad & Curran, 2016*). Specifically, in all the simulation studies, the prior distributions of the

parameters were set to a normal distribution with a large variance,

$$\theta = (\beta_0, \beta_1, \beta_2, \beta_3) \sim N(0, 10^2 I_4)$$

$$\gamma \sim N(0, 10^2)$$

where $I_4$ is the identity matrix of size 4, which is a $4 \times 4$ square matrix with ones on the main diagonal and zeros elsewhere.

Three criteria were used to evaluate the extent to which the empirical coverage of the confidence interval matched the nominal 95% level. The first was the rank of the deviation from the nominal level, which was obtained by ranking the absolute value of the difference between the coverage rate and 95%. The second was the percentage of times that the interval covered the true value, which fell within the range of 93.6–96.4% (due to the 1,000 samples). The third was the percentage of missed data in either direction. The left miscoverage rate was the proportion of times the lower limits were larger than the true value, while the right miscoverage rate was the proportion of times the upper limits were less than the true value. A balanced miscoverage indicated that the confidence interval was neither too wide nor too narrow. From the usual sample-size formula for estimating proportions, if the true coverage rate for a particular type of confidence interval (or credible interval) was 95%, then the probability of obtaining coverage between 92.5% and 97.5% would be approximately 0.95. Thus, an advertised 95% confidence interval (or credible interval) should miss approximately 2.5% from each side; *i.e.,* the left/right miscoverage rates should be 2.5%.

## RESULTS OF THE SIMULATION STUDY

With respect to the balanced design with 300 cases and controls, for each scenario and each method, there were 1,000 point estimates of the interaction measures with considerable variation. However, for each interaction measure, the mean and median of the point estimates were very close to the corresponding true values. For instance, in scenario A1, the true value of the RERI was 12. The 1,000-point estimates of delta for the RERI ranged from −2.83 to 32.91; the mean and median values were 13.21 and 11.95, respectively, and the standard deviation was 7.62.

The sampling distributions of those point estimates, estimated from the same datasets, were almost the same among the three methods. This was consistent with expectations. Figure 1 shows the sampling distribution of each interaction measure in scenarios A1, A3 and A5, corresponding to strong synergy, no interaction, and strong antagonism, respectively. The RERI seemed to follow a normal distribution with a mean of 0 when there was no interaction but a mildly right-skewed distribution when there was synergy and a mildly left-skewed distribution when there was antagonism. For AP, the distributions were skewed to the left in all scenarios except for antagonism. For S, the distributions were skewed to the right in every scenario.

Table 2 summarizes the coverage performance of the three estimation methods for the RERI. Due to the skewed distribution, when using the median as the estimate, all the values

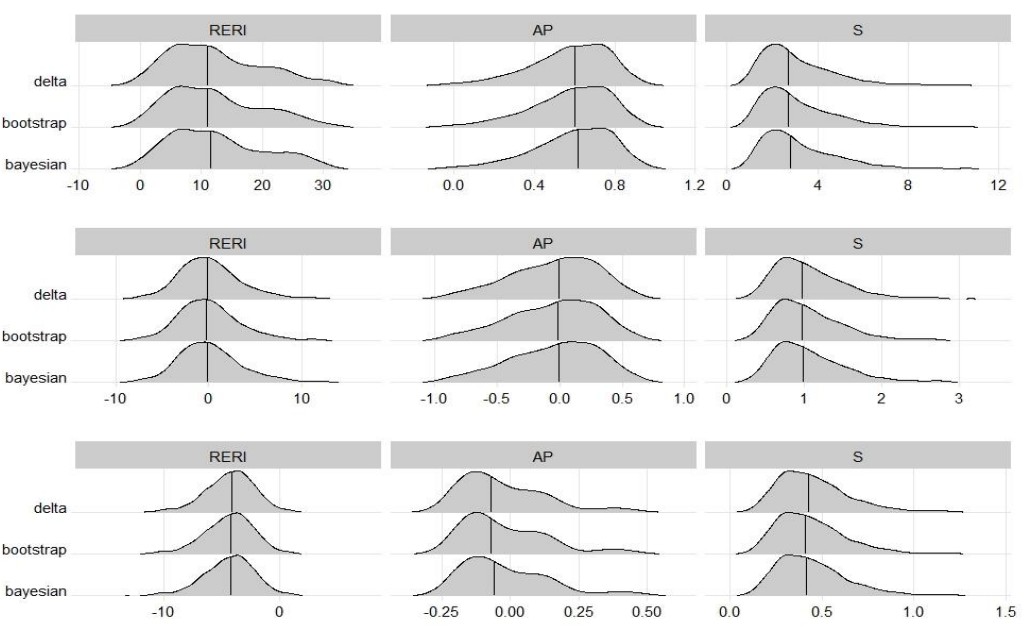

**Figure 1  Sampling distributions of interaction measures, which were estimated from the scenarios A1 (top), A3 (middle) and A5 (bottom) in the balanced design simulation study.** The lines reveal the median of the distribution which were very close to the true value of RERI, AP and S, respectively.

were close to the true value of the RERI. For the delta, bootstrap, and Bayesian intervals, the average ranks of the deviation were 2.75, 1.50 and 1.65, respectively. This shows that, over a wide range of scenarios, the bootstrap method most often yields a coverage rate closest to 95%, followed by the Bayesian method. Notably, the rank would be equal to 3 if the coverage rate was the farthest from 95% in every scenario, which indicates that the delta intervals least accurately match the nominal value of 95%.

For delta intervals, the coverage rate fit into the target range of 93.6–96.4% in 4 of the 20 scenarios. Clearly, the coverage rate tended to be low when there was synergy and high when there was antagonism. In particular, in scenarios with synergy, delta intervals covered the true value of the RERI by lying too far to the right, which resulted in low right miscoverage and high left miscoverage. However, in scenarios with antagonism, the delta intervals may be too wide, resulting in a high coverage rate and low miscoverage. In contrast, the Bayesian coverage was within the target range in 18 of the 20 scenarios. Furthermore, the left/right miscoverages were approximately 2.5%, which indicates that the Bayesian intervals were more evenly balanced. The bootstrap intervals were similar to those of the Bayesian method.

Table 3 summarizes the coverage performance of the three estimation methods for AP. For the delta, bootstrap, and Bayesian intervals, the average ranks of the deviation were 3, 1.45 and 1.4, respectively. The coverage for delta intervals was in the target range for only two of the 20 scenarios, while that for the bootstrap and Bayesian intervals was in the target range for 19 of the 20 scenarios. For the delta interval, the miscoverage was lop-sided, with high right miscoverage. It tended to achieve an excessively high lower

**Table 2  Coverage properties of the 95% two-sided confidence/credible intervals for RERI in balanced design study.**

| | Delta | | | | Bootstrap | | | | Bayesian | | | |
|---|---|---|---|---|---|---|---|---|---|---|---|---|
| RERI | Estimate | Left | Cover | Right | Estimate | Left | Cover | Right | Estimate | Left | Cover | Right |
| 12.00 | 11.95 | 7.3 | 92.7 | 0.0 | 12.30 | 1.3 | 94.4 | 4.3 | 12.93 | 2.1 | 93.8 | 4.1 |
| 4.00 | 3.82 | 5.2 | 94.8 | 0.0 | 3.78 | 2.1 | 95.3 | 2.6 | 4.07 | 2.8 | 94.8 | 2.4 |
| 0.00 | −0.17 | 3.2 | 96.8 | 0.0 | −0.24 | 2.9 | 94.8 | 2.3 | −0.13 | 3.1 | 94.9 | 2.0 |
| −2.00 | −2.11 | 1.7 | 98.2 | 0.1 | −2.17 | 3.1 | 94.8 | 2.1 | −2.15 | 3.6 | 94.6 | 1.8 |
| −4.00 | −4.11 | 0.7 | 97.9 | 1.4 | −4.18 | 3.3 | 94.7 | 2.0 | −4.22 | 3.4 | 94.5 | 2.1 |
| 9.00 | 9.03 | 8.8 | 91.2 | 0.0 | 9.21 | 2.9 | 92.5 | 4.6 | 9.63 | 3.3 | 92.8 | 3.9 |
| 3.00 | 2.81 | 6.4 | 93.6 | 0.0 | 2.83 | 2.8 | 94.4 | 2.8 | 2.94 | 3.5 | 94.3 | 2.2 |
| 0.00 | −0.09 | 3.4 | 96.6 | 0.0 | −0.10 | 3.4 | 94.1 | 2.5 | −0.08 | 3.8 | 94.1 | 2.1 |
| −1.50 | −1.52 | 1.4 | 98.4 | 0.2 | −1.55 | 3.4 | 94.7 | 1.9 | −1.56 | 3.0 | 95.0 | 2.0 |
| −3.00 | −3.03 | 0.7 | 97.2 | 2.1 | −3.05 | 3.7 | 94.0 | 2.3 | −3.11 | 4.0 | 93.7 | 2.3 |
| 8.25 | 8.44 | 7.5 | 92.5 | 0.0 | 8.48 | 1.7 | 94.0 | 4.3 | 8.94 | 2.4 | 93.8 | 3.8 |
| 2.75 | 2.70 | 4.6 | 95.4 | 0.0 | 2.67 | 2.4 | 95.0 | 2.6 | 2.82 | 3.5 | 94.3 | 2.2 |
| 0.00 | −0.06 | 1.7 | 98.3 | 0.0 | −0.09 | 2.7 | 95.3 | 2.0 | −0.07 | 3.6 | 94.3 | 2.1 |
| −1.38 | −1.35 | 0.5 | 99.1 | 0.4 | −1.39 | 2.7 | 95.4 | 1.9 | −1.38 | 3.2 | 94.7 | 2.1 |
| −2.75 | −2.80 | 0.4 | 96.9 | 2.7 | −2.85 | 3.3 | 94.8 | 1.9 | −2.88 | 3.5 | 94.1 | 2.4 |
| 5.25 | 5.13 | 7.5 | 92.5 | 0.0 | 5.17 | 2.7 | 94.4 | 2.9 | 5.37 | 3.6 | 93.9 | 2.5 |
| 1.75 | 1.70 | 6.2 | 93.8 | 0.0 | 1.66 | 3.6 | 93.5 | 2.9 | 1.70 | 4.6 | 92.9 | 2.5 |
| 0.00 | −0.04 | 3.1 | 96.9 | 0.0 | −0.06 | 3.1 | 95.2 | 1.7 | −0.06 | 3.4 | 95.1 | 1.5 |
| −0.88 | −0.87 | 1.9 | 97.9 | 0.2 | −0.89 | 3.8 | 94.6 | 1.6 | −0.91 | 3.7 | 94.7 | 1.6 |
| −1.75 | −1.80 | 0.9 | 96.7 | 2.4 | −1.83 | 4.4 | 93.8 | 1.8 | −1.84 | 4.7 | 93.9 | 1.4 |

confidence limit, which would probably lead to false positive results. The Bayesian interval had better coverage, but the miscoverage appeared to be slightly unbalanced, as in the case of the bootstrap interval. Nevertheless, from a practical standpoint, this miscoverage is reasonable.

Table 4 summarizes the coverage performance of the three estimation methods for S. For the delta, bootstrap, and Bayesian intervals, the average ranks of the deviation were 2.35, 1.60 and 1.85, respectively, and the numbers of points within the target range were 15, 19 and 18, respectively. Notably, the imbalance in miscoverage was slightly alleviated for the three estimation methods.

Further simulation results with imbalanced designs or confounding variables are reported in the supplementary material (Tables S1–S6) for the interaction measures RERI, AP and S. In general, the performances of both studies exhibited similar statistical behavior as that of the balanced design. Specifically, for AP, the coverage rates of the delta interval were especially low, and the worst coverage rate was as low as 87.6%. However, the coverage rates of the bootstrap and Bayesian intervals were mostly slightly less than 95%. In addition, the bootstrap and Bayesian intervals achieved balanced miscoverage, but the delta interval did not. In summary, the simulations demonstrate that, for three interaction measures, the bootstrap and Bayesian methods performed satisfactorily, while the delta method did

**Table 3  Coverage properties of the 95% two-sided confidence intervals for AP in balanced design study.**

| | Delta | | | | Bootstrap | | | | Bayesian | | | |
|---|---|---|---|---|---|---|---|---|---|---|---|---|
| AP | Estimate | Left | Cover | Right | Estimate | Left | Cover | Right | Estimate | Left | Cover | Right |
| 0.60 | 0.60 | 0.0 | 92.2 | 7.8 | 0.60 | 1.5 | 94.5 | 4.0 | 0.61 | 2.7 | 93.7 | 3.6 |
| 0.33 | 0.33 | 0.2 | 93.2 | 6.6 | 0.32 | 2.2 | 95.3 | 2.5 | 0.34 | 2.7 | 95 | 2.3 |
| 0.00 | −0.02 | 0.6 | 93.3 | 6.1 | −0.03 | 2.9 | 94.8 | 2.3 | −0.02 | 3.1 | 94.9 | 2.0 |
| −0.33 | −0.37 | 0.6 | 93.7 | 5.7 | −0.38 | 3.1 | 94.8 | 2.1 | −0.37 | 3.3 | 94.8 | 1.9 |
| −1.00 | −1.04 | 0.3 | 93.1 | 6.6 | −1.08 | 3.0 | 94.7 | 2.3 | −1.06 | 3.4 | 94.5 | 2.1 |
| 0.60 | 0.60 | 0.6 | 91.8 | 7.6 | 0.61 | 3.0 | 93.2 | 3.8 | 0.62 | 3.5 | 93.2 | 3.3 |
| 0.33 | 0.33 | 0.7 | 92.5 | 6.8 | 0.32 | 3.1 | 94.1 | 2.8 | 0.33 | 3.4 | 94.4 | 2.2 |
| 0.00 | −0.01 | 0.6 | 93.5 | 5.9 | −0.02 | 3.4 | 94.1 | 2.5 | −0.01 | 3.8 | 94.1 | 2.1 |
| −0.33 | −0.34 | 0.4 | 92.5 | 7.1 | −0.36 | 3.1 | 95.0 | 1.9 | −0.35 | 3.1 | 94.8 | 2.1 |
| −1.00 | −1.03 | 0.1 | 92.4 | 7.5 | −1.04 | 3.7 | 93.6 | 2.7 | −1.05 | 3.4 | 94.0 | 2.6 |
| 0.60 | 0.60 | 0.1 | 91.5 | 8.4 | 0.61 | 1.8 | 94.6 | 3.6 | 0.62 | 2.8 | 93.9 | 3.3 |
| 0.33 | 0.34 | 0.1 | 93.8 | 6.1 | 0.33 | 2.4 | 95.0 | 2.6 | 0.34 | 3.0 | 94.4 | 2.6 |
| 0.00 | −0.01 | 0.2 | 92.7 | 7.1 | −0.02 | 2.8 | 95.2 | 2.0 | −0.01 | 3.6 | 94.3 | 2.1 |
| −0.33 | −0.34 | 0.2 | 92.7 | 7.1 | −0.35 | 2.9 | 95.6 | 1.5 | −0.35 | 3.0 | 95.3 | 1.7 |
| −1.00 | −1.00 | 0.0 | 93.1 | 6.9 | −1.04 | 3.2 | 95.0 | 1.8 | −1.05 | 3.3 | 94.8 | 1.9 |
| 0.60 | 0.60 | 1.0 | 92.4 | 6.6 | 0.60 | 3.2 | 94.9 | 1.9 | 0.60 | 3.8 | 94.2 | 2.0 |
| 0.33 | 0.33 | 0.8 | 92.4 | 6.8 | 0.33 | 3.5 | 94.1 | 2.4 | 0.32 | 3.8 | 94.2 | 2.0 |
| 0.00 | −0.01 | 1.1 | 91.3 | 7.6 | −0.02 | 3.1 | 95.2 | 1.7 | −0.02 | 3.4 | 95.1 | 1.5 |
| −0.33 | −0.33 | 0.8 | 91.3 | 7.9 | −0.35 | 3.1 | 95.5 | 1.4 | −0.34 | 3.3 | 95.1 | 1.6 |
| −1.00 | −1.01 | 1.0 | 90.3 | 8.7 | −1.04 | 3.7 | 94.1 | 2.2 | −1.02 | 3.2 | 94.8 | 2.0 |

not. The difference between the bootstrap and Bayesian methods was minimal and can be considered negligible.

# EXAMPLE

This example came from a case-control study among male veterans younger than 60 years concerning smoking and alcohol use in relation to oral cancer (*Rothman & Keller, 1972*). The data are presented in Table 5. Several authors have used various methods to calculate confidence intervals (CIs) for the RERI. Specifically, based on Hosmer and Lemeshow's delta method (*Hosmer & Lemeshow, 1992*), the point estimate of the RERI was 3.74, and the asymptotic 95% confidence interval was (−1.85 to 9.33). Based on Zou's variance estimates recovery method (MOVER) (*Zou, 2008*), the 95% confidence interval was (−11.41 to 21.84). Based on Richardson and Kaufman's profile likelihood method (*Richardson & Kaufman, 2009*), the 95% confidence interval was (−3.29 to 17.21). The 95% confidence interval of the percentile bootstrap method proposed by *Assmann et al. (1996)* was (−10.77 to 1.6 × $10^7$), which was too wide due to the sparse cells in some bootstrapped samples. In addition, *Chu, Nie & Cole (2011)* proposed a Bayesian method for linear additive odds ratio models and reported that the posterior medians of the RERI with 95% high probability density and credible intervals were 3.00 (−2.27 to 8.90), 4.05 (−2.54 to 12.65) and 4.75 (−2.63 to 15.60), corresponding to different prior distributions.

**Table 4  Coverage properties of the 95% two-sided confidence intervals for S in balanced design study.**

|  | Delta | | | | Bootstrap | | | | Bayesian | | | |
|---|---|---|---|---|---|---|---|---|---|---|---|---|
| S | Estimate | Left | Cover | Right | Estimate | Left | Cover | Right | Estimate | Left | Cover | Right |
| 2.71 | 2.70 | 3.1 | 96.8 | 0.1 | 2.73 | 1.5 | 94.7 | 3.8 | 2.84 | 2.6 | 93.5 | 3.9 |
| 1.57 | 1.56 | 2.8 | 95.8 | 1.4 | 1.56 | 2.3 | 95.2 | 2.5 | 1.59 | 2.5 | 95.2 | 2.3 |
| 1.00 | 0.98 | 3.0 | 95.8 | 1.2 | 0.97 | 2.9 | 94.8 | 2.3 | 0.98 | 3.1 | 94.9 | 2.0 |
| 0.71 | 0.70 | 2.6 | 95.8 | 1.6 | 0.69 | 3.2 | 94.7 | 2.1 | 0.69 | 3.4 | 94.7 | 1.9 |
| 0.43 | 0.42 | 2.1 | 95.5 | 2.4 | 0.41 | 3.1 | 94.5 | 2.4 | 0.42 | 3.5 | 94.4 | 2.1 |
| 2.80 | 2.84 | 3.8 | 95.8 | 0.4 | 2.89 | 3.0 | 93.1 | 3.9 | 2.93 | 3.7 | 93.0 | 3.3 |
| 1.60 | 1.57 | 3.6 | 95.1 | 1.3 | 1.58 | 3.2 | 93.9 | 2.9 | 1.59 | 3.4 | 94.3 | 2.3 |
| 1.00 | 0.98 | 3.3 | 94.8 | 1.9 | 0.97 | 3.4 | 94.1 | 2.5 | 0.99 | 3.8 | 94.1 | 2.1 |
| 0.70 | 0.69 | 2.1 | 96.1 | 1.8 | 0.68 | 2.9 | 95.2 | 1.9 | 0.69 | 3.2 | 94.7 | 2.1 |
| 0.40 | 0.39 | 1.0 | 96.2 | 2.8 | 0.38 | 3.4 | 94.1 | 2.5 | 0.39 | 2.9 | 94.5 | 2.6 |
| 2.83 | 2.90 | 3.3 | 95.6 | 1.1 | 2.97 | 2.0 | 94.8 | 3.2 | 3.04 | 2.8 | 94.1 | 3.1 |
| 1.61 | 1.61 | 2.8 | 95.9 | 1.3 | 1.62 | 2.3 | 95 | 2.7 | 1.63 | 2.9 | 94.6 | 2.5 |
| 1.00 | 0.98 | 2.5 | 95.8 | 1.7 | 0.98 | 2.8 | 95.2 | 2.0 | 0.99 | 3.6 | 94.3 | 2.1 |
| 0.69 | 0.69 | 1.9 | 96.6 | 1.5 | 0.68 | 2.8 | 95.6 | 1.6 | 0.69 | 3.0 | 95.3 | 1.7 |
| 0.39 | 0.39 | 0.7 | 97.2 | 2.1 | 0.38 | 2.8 | 95.6 | 1.6 | 0.38 | 3.3 | 95.0 | 1.7 |
| 3.10 | 3.11 | 4.6 | 94.6 | 0.8 | 3.15 | 2.9 | 95.6 | 1.5 | 3.15 | 3.9 | 94.5 | 1.6 |
| 1.70 | 1.68 | 3.5 | 95.2 | 1.3 | 1.70 | 3.5 | 94.6 | 1.9 | 1.68 | 3.7 | 94.5 | 1.8 |
| 1.00 | 0.98 | 2.4 | 96.4 | 1.2 | 0.97 | 3.1 | 95.4 | 1.5 | 0.98 | 3.4 | 95.1 | 1.5 |
| 0.65 | 0.65 | 0.7 | 97.8 | 1.5 | 0.64 | 2.9 | 95.4 | 1.7 | 0.64 | 2.9 | 95.6 | 1.5 |
| 0.30 | 0.29 | 0.0 | 96.7 | 3.3 | 0.28 | 2.7 | 95.2 | 2.1 | 0.29 | 1.9 | 95.5 | 2.6 |

**Table 5  Distribution of exposures among cases and controls in the oral cancer example[a].**

|  | Neither | Smoking only | Alcohol only | Smoking and alcohol |
|---|---|---|---|---|
| Cases | 3 | 8 | 6 | 225 |
| Controls | 20 | 18 | 12 | 166 |

**Notes.**
[a] data were from the example presented by Rothman and Keller.

For the data at hand, using the proposed Bayesian method, the point estimates with 95% credible intervals for REIR, AP and S are shown in Table 6, along with the confidence intervals of the delta and bootstrap methods. The three interaction measures of all methods suggested synergy, but the 95% intervals were broad and not significant. For the RERI, the Bayesian method provided a reasonable 95% credible interval ($-3.93$ to $19.59$). For AP, owing to the left-skewed distribution, the Bayesian interval was positioned further left than the delta interval was. However, for S, owing to the right-skewed distribution, the Bayesian interval was positioned further right than the delta interval was. Figure 2 shows the distributions of interactions in this example. Furthermore, the features that appeared in the simulation studies seemed to be maintained.

Notably, the overall performance of the bootstrap method was comparable to that of the Bayesian method in all the simulation studies. However, in this example, regardless of the

**Table 6  Point estimates with 95% confidence limits for interaction effects in oral cancer example.**

|  | Delta | Bootstrap | Bayesian |
|---|---|---|---|
| RERI | 3.74 (−1.84 to 9.32) | 3.51 (−4.86 to 5.78 ×10⁶) | 3.68 (−3.93 to 19.59) |
| AP | 0.41 (−0.07 to 0.90) | 0.38 (−0.32 to 0.83) | 0.38 (−0.34 to 0.72) |
| S | 1.87 (0.65 to 5.42) | 1.69 (0.66 to 11.24) | 1.71 (0.72 to 5.78) |

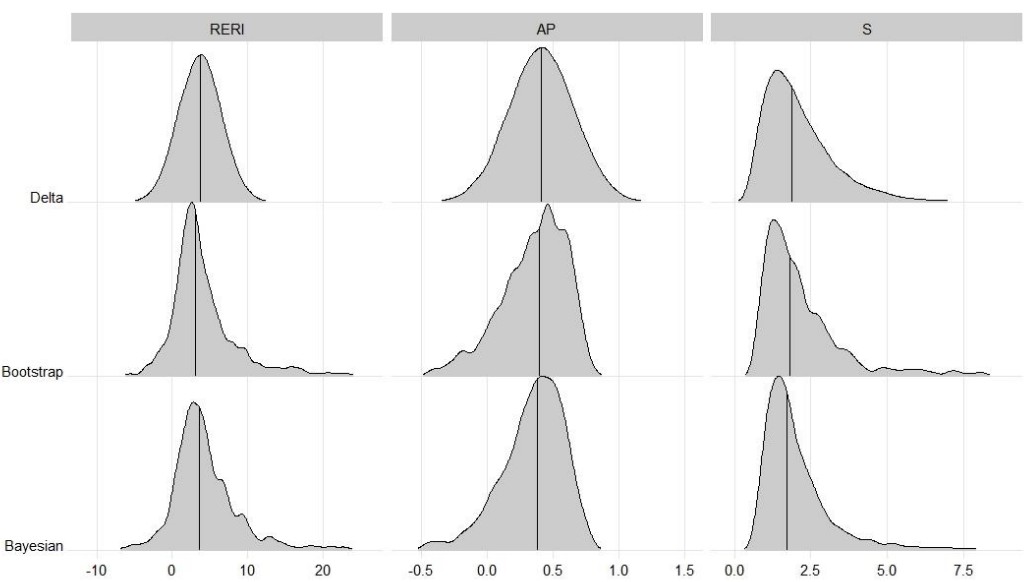

**Figure 2  Distributions of interaction in oral cancer example.** The distributions for Delta were based on the normal distribution (RERI and AP) or lognormal distribution (S) with estimated mean and standard error, respectively.

three interaction measures, the intervals in the Bayesian method were narrower than those in the bootstrap method, which may indicate that there is less variance in the Bayesian method. Notably, the upper limits of the intervals obtained *via* the bootstrap method were greater than those obtained *via* the Bayesian method, especially for the RERI. This indicates that the percentile bootstrap may perform poorly when data with a small sample size exist in some groups, but the proposed Bayesian approach is free from this predicament.

## DISCUSSION

It is very important to assess whether an additive interaction exists between two factors during the interpretation of epidemiologic data because this information is relevant to disease prevention and intervention. In general, in addition to hypothesis tests, confidence intervals are used in the assessment of statistical interactions (*Greenland et al., 2016*). There are many methods for constructing confidence intervals for RERI, such as the delta method (*Hosmer & Lemeshow, 1992*), bootstrap method (*Assmann et al., 1996*), variance estimate recovery approach (*Zou, 2008*), and profile likelihood approach (*Richardson & Kaufman, 2009*), based on the theories of classical frequentists. However, due to the complexity of

the formulas, there is little information available for AP or S. An alternative method is the Bayesian approach, which uses a credible interval rather than a confidence interval. *Chu, Nie & Cole (2011)* proposed a Bayesian method for linear additive odds ratio models to construct the credible interval of the RERI but not that of the AP or S. This article proposes a Bayesian method to estimate all the additive interaction measures, RERI, AP and S, along with their 95% credible intervals. In contrast to the method of *Chu, Nie & Cole (2011)*, in which the posterior distribution is a step function, this method does not require inequality constraints on the parameters, which may expand the sample space of the parameter and lead to higher efficiency when MCMC is applied.

The simulation studies show that the distributions of the additive interaction measures are skewed, especially those of AP and S. Therefore, the delta method, whose Wald-type confidence interval is symmetric, performs poorly in terms of the empirical coverage of the confidence interval or the percentage of times missed in either direction. By recovering the variances, the method proposed by *Zou (2008)* may alleviate this situation. However, as Wald-type confidence intervals are based on asymptotic variance, they may be misleading in nonmultiplicative models (*Moolgavkar & Venzon, 1987*).

Compared with those of the delta method, the Bayesian credible intervals proposed here and the bootstrap confidence intervals, which do not rely on asymptotic approximations, are more likely to catch on each side of the true value, and their coverage is close to the nominal level of 95%. The bootstrap confidence intervals, as proposed by Assmann et al., are preferred in practice (*Töpper et al., 2018*). However, the bootstrap method has several limitations. *DiCiccio & Efron (1996)* cautioned that a bootstrap sample may depend on a distribution with difficult computations in nonmultiplicative models. In addition, as observed in the example, the confidence interval could come to nothing when the sample is insufficient (*Chernick, 1999*). However, Bayesian credible intervals have minimal impact and work well.

Some authors (*Rothman, Greenland & Lash, 2008*; *Knol et al., 2011*) have noted that additive measures are suitable only for risk factors. If there is a preventive factor, it should be recoded to a risk factor before calculating these measures. To determine where it is a preventive factor and how to recode it, the logistic regression model needs to be applied at least twice. When this strategy is applied to the bootstrap method, each bootstrap sample needs to be recoded separately rather than used for the bootstrap samples. This approach significantly increases the calculation time. However, in the Bayesian framework, once a sample of the posterior distribution is obtained, one can apply this sample to any function (*Bolstad & Curran, 2016*). Therefore, the posterior distributions of the additive measures can be estimated when the coefficients $\theta = (\beta_0, \beta_1, \beta_2, \beta_3)$ are obtained by the Bayesian method. This makes the model presented here highly flexible. As shown in Appendix S1A, the additive measures are computed directly after parameter transformation, but the computation time can be ignored. Moreover, one can adjust the confounding variable X in various forms in logistic regression, such as the linear or spline function of X.

The frequentist and Bayesian methods appear to be superficially opposite, but they can be considered complementary in practice (*Chu, Nie & Cole, 2011*). When using a weak prior distribution in Bayesian methods, the inference is often consistent with that of

frequentists. However, the Bayesian method is attractive because proper prior distributions can be determined from past information, such as subject-matter knowledge or previous experiments (*Bolstad & Curran, 2016*). Furthermore, the credible interval is constructed directly from the highest probability density of the posterior samples, and it can capture the uncertainty of the parameter values without having to rely on asymptotic approximations. This approach can often yield better results, especially for small samples, as shown in the example. In addition, Bayesian methods have a range of other advantages, including convenient application to other indicators related to interactions (*e.g.*, PRISM (*Lee, 2013*; *Lin & Lee, 2016*), CPWs (*Lee & Wu, 2023*) and the ability to explore disease attribution to multiple exposures and their interactions.

In this article, a Bayesian method for the estimation of additive measures is proposed. By rearranging the posterior sample, it works for both risk factors and preventive factors. The simulation studies and example results show that the Bayesian method is a competitive alternative to other methods.

### Funding
This research was funded by the Central Government-Led Local Science and Technology Development Special Project (No. 2019L3006), the Central Government-Led Local Science and Technology Development Special Project (No. 2020L3009), the Scientific and Technological Innovation Joint Capital Projects of Fujian Province (No. 2020Y9018), the Natural Science Foundation of Fujian Province (No. 2021J01726), and the Natural Science Foundation of Fujian Province (No. 2021J01733). The funders had no role in study design, data collection and analysis, decision to publish, or preparation of the manuscript.

### Grant Disclosures
The following grant information was disclosed by the authors:
Central Government-Led Local Science and Technology Development Special Project: 2019L3006, 2020L3009.
Scientific and Technological Innovation Joint Capital Projects of Fujian Province: 2020Y9018.
Natural Science Foundation of Fujian Province: 2021J01726, 2021J01733.

### Competing Interests
The authors declare there are no competing interests.

### Author Contributions
- Shaowei Lin conceived and designed the experiments, performed the experiments, analyzed the data, prepared figures and/or tables, authored or reviewed drafts of the article, and approved the final draft.
- Chanchan Hu performed the experiments, analyzed the data, prepared figures and/or tables, and approved the final draft.

- Zhifeng Lin performed the experiments, analyzed the data, prepared figures and/or tables, and approved the final draft.
- Zhijian Hu conceived and designed the experiments, authored or reviewed drafts of the article, and approved the final draft.

## Data Availability

The raw measurements are available in the Supplementary Files.

## Supplemental Information

Supplemental information for this article can be found online at http://dx.doi.org/10.7717/peerj.17128#supplemental-information.

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
