# Peer review of "Bayesian estimation of the measurement of interactions in epidemiological studies"

_PeerJ, doi:10.7717/peerj.17128_

## Round 0.1 · original submission · Major Revisions

Thank you for your manuscript. As you will see, two reviewers have assessed your manuscript and made relatively favourable comments. I will ask you to address each of their comments in your rebuttal, for each one indicating either how you have changed your manuscript in response (with such changes tracked) or why you believe no changes are required. I’ll make some additional comments of my own below, which I will ask you to address in the same way.

Line 38: I think that this could be worded more clearly as each factor’s value has an effect on the other factor’s (or factors’) effect rather than just on one another.

Lines 94–100: I wondered why you didn’t also consider AP* (RERI/(RR_11-1)) here which Rothman describes as being potentially of interest also. I only have Rothman’s 4th edition on hand but I thought that this had been covered in earlier editions.

Lines 179–180 : While you referred to “equal-tailed interval” back on Line 88, I think the same should be done here. This isn’t the only option for a 95% credible interval, of course.

Line 180: I found the discussion of hypothesis testing to be contrary to the usual intentions around Bayesian analyses. I wonder if, assuming you agree, this could be reframed around the posterior distribution and probabilistic statements based on this.

Line 256: Perhaps “was 5% overall” as you’ve already noted the 2.5% per tail?

Reviewer 1 ·

Basic reporting

This study focused on the assessment of interaction measurement in epidemiological research. The authors recommended the use of Bayesian logistic regression to determine the credible intervals for additive interaction indices, including the relative excess risk due to interaction (RERI), attributable proportion due to interaction (AP), and synergy index (S). The paper is well-structured and articulated. Below are some comments for consideration.
(1) The PRISM (Peril Ratio Index of Synergy based on Multiplicativity) is an interaction measure derived from the sufficient component cause model. Its estimation can be achieved through complementary log regression. The authors might consider discussing the feasibility of applying a Bayesian approach to this index.
(2) The authors might consider discussing interval estimation in the context of the monotonicity assumption. Specifically, when the effects of both factors are known to be unidirectional, either both being risk-enhancing with positive interaction or both being protective with negative interaction.
(3) The authors could delve into the pertinent topic of disease attribution to multiple exposures and their interactions, especially in the context of global burden of disease studies.

References
1. Assessing causal mechanistic interactions: a peril ratio index of synergy based on multiplicativity. PLoS ONE 2013;8(6):e67424.
2. Complementary log regression for sufficient cause modeling of epidemiologic data. Sci Rep 2016;6, 39023; doi: 10.1038/srep39023.
3. Disease attribution to multiple exposures using aggregate data from multiple sources. J Epidemiol 2023; https://doi.org/10.2188/jea.JE20210084.

Experimental design

nothing to add.

Validity of the findings

nothing to add.

Additional comments

nothing to add.

Reviewer 2 ·

Basic reporting

The authors present a simulation study examining three methods of obtaining confidence intervals for logistic regression models that include an interaction. The basic premise is sound, and the paper has some merit for educational purposes although the results will not be surprising to anyone that has worked with these types of models before. The fact that complete R code to replicate the simulations is provided is also commendable. The article also has a generally high standard of english. However, there is no citation for R, Stan or JAGS despite using R (and citing MCMCpack) and mentioning Stan and JAGS in the text. Please cite this software appropriately (partiocularly R, as you have used it for your work).

Experimental design

The experimental design (simulation study) is sound, although I would like to see some indication of the typical time taken for each estimation method (the historical advantage of approximations such as Wald was computational tractability).

It may also be interesting to see how the methods compare for estimating the main effects (either one or both) as OR. Could this be added to the paper, perhaps with full details in the appendix?

It is also a shame that the authors do not set a PRNG seed in the R code - this would have made the results completely replicable, rather than being replicable to within a numerical approximation due to random variability in the data simulation and bootstrap/MCMC procedures.

Validity of the findings

The findings seem to be valid, and the provision of the R code to repeat the simulation study gives high confidence that the study was performed correctly.

Additional comments

I think the section that discusses methods of confidence interval estimation would benefit from a brief discussion of profile likelihood methods, such as those used by TMB.

The section discussing the difference between relative risks and odds ratios could use some work; for example why are OR good approximations to RR for rare outcomes, and why is the coding of the reference category important? It would also be useful to include an example of how RR can be calculated from OR by correcting for the intercept value (particularly when bootstrapping or using MCMC).

---

## Round 0.2 · Minor Revisions

Thank you for your revisions and responses. As you can see, Reviewer #1 has no further comments and Reviewer #2 has only comments on the citations. I’ll ask you to address those along with some minor comments from me (below). Providing these are all responded to/addressed appropriately, I would hope to be able to quickly accept a revised version of your manuscript.

1. The text would be improved by careful editing by a fluent English speaker. The meaning is clear throughout for me, but some of the phrasing could be made more natural and some language issues need to be corrected.

2. There is a spurious carriage return between Lines 123 and 124.

3. Line 144’s “so do” might have been intended to be “including”?

4. There is a missing space on Line 285 (“0when”).

5. Table 2’s caption could read “confidence/credible intervals” as both are included in the table. The heading row should read “Bayesian” (capitalized “b”, c.f. Table 6). See also Tables 3 and 4 for the same points and Table 6 for the caption point only.

Reviewer 1 ·

Basic reporting

Satisfactory

Experimental design

Satisfactory

Validity of the findings

Valid findings

Additional comments

The authors have updated the paper, incorporating my suggestions.

Reviewer 2 ·

Basic reporting

Thanks for considering my comments - I think your changes have improved the paper. However, the citations for JAGS and R are incorrect. JAGS is typically cited as shown at https://www.scirp.org/reference/referencespapers?referenceid=3070812 and R itself shows you how to cite it using the citation() function for both R i.e. just citation() and packages i.e. citation("MCMCpack") - please correct these. And for Stan you need to replace the text YEAR and VERSION with the year and version that you actually used.

Experimental design

--

Validity of the findings

--

Additional comments

--

---

## Round 0.3 · accepted · Accept

Thank you for your revised manuscript, which I am delighted to be able to accept. Well done on your revisions. I look forward to seeing your work published and the discussion I hope it will generate.

I will make suggestions of possible edits to wording or formatting, which you could incorporate during the proofing process. None of these reflect the content of your work, but they are intended to help readers of your article once published.

Line 40: The addition of a space before references might make these easier for readers.

Line 140: The estimated variance seems lower compared to the surrounding text. There might be other instances of this that also need correcting.

Line 147: Perhaps “any statistic using random sampling, including the measures of additive interaction” rather than “any statistic using random sampling, as do the measures of additive interaction”.

Line 155: The alpha here seems raised compared to the surrounding text and larger than the alpha on Line 157. As above, check for other instances. (And note the seemingly lowered Z just before this.)

Line 193: Is there a space before Theta here?

Line 196: While I won’t point out all instances of raised/lowered equations, this one is particularly conspicuous for me, as are the instances on Lines 216 and 238. See also Line 261.

Line 218: Perhaps “An R…” rather than “The R…” as your function isn’t part of Base R and alternative implementations are possible.

Line 227: Again, this isn’t the only instance, but the spacing around this line seems unnecessarily large (see also Lines 124–127, 154, 212, etc.).

Line 250: Please give the version number for MCMCpack.